# Latest Performance Improvement Strategies and Techniques Used in 5G Antenna Designing Technology, a Comprehensive Study

**DOI:** 10.3390/mi13050717

**Published:** 2022-04-30

**Authors:** Iftikhar Ahmad, Wenhao Tan, Qasim Ali, Houjun Sun

**Affiliations:** Beijing Key Laboratory of Millimeter Wave and Terahertz Techniques, School of Information and Electronics, Beijing Institute of Technology, Beijing 100081, China; iftikhar.roghani@bit.edu.cn (I.A.); tanwenhao@bit.edu.cn (W.T.); qasimali@bit.edu.cn (Q.A.)

**Keywords:** 5G, mmWave, SIW, hybrid feeding, SIC, wide bandwidth, ECC, stable high gain, isolation, MEG, DIG, SAR, SSL, metamaterials (MMTs), DRAs

## Abstract

In the recent era, fifth-generation technology (5G) has not been fully implemented in the realm of wireless communication. To have excellent accessible bandwidth feasibility, and in order to achieve the aims of 5G standards, such as higher data rates and ultrahigh-definition video streaming, the millimeter wave (mmWave) band must be employed. Services with minimal latency and many other features are feasible only in the mmWave spectrum. To avoid numerous communication complexities such as high connection losses, short wavelength, and restricted bandwidth, as well as path-loss challenges in the mmWave range, an antenna with wide bandwidth, high gain, narrow steerable beam, high isolation, low side-lobe levels, and multiband features is required to alleviate these difficulties and meet 5G communication standards. To overcome these challenges, specific strategies and techniques should be employed in the traditional antenna designing procedure to excellently improve the performance of the antenna in terms of bandwidth, gain, and efficiency and to reduce the mutual coupling effect between the closely colocated antenna elements in MIMOs and arrays. The researchers reported on a variety of bandwidth and gain improvement approaches. To gain broader coverage, traditional antenna design techniques must be modified. In this study, the latest state-of-the-art work is reviewed, such as the role of the metamaterials (MMTs), parasitic patches, hybrid feeding, EBG structure, impact of the slots with different geometrical shapes in the radiator to achieve the goal of wide bandwidth, boosted gain, reduced side-lobes level, as well as stable radiation properties. Mutual coupling reduction techniques are also briefly reported. The role of reconfigurability is focused on in this study, and at the end, the future challenges in the field of antenna design and possible remedies to such issues are reviewed.

## 1. Introduction

In mobile communication, 5G is the latest technical advancement [1,2,3,4,5,6,7,8,9,10]. Facilitating users with delivery of high data rates, HD video broadcasting with ultrahigh speed, flawless traffic capacity, as well as an improved network and spectrum efficiency or efficient frequency sweep [2,3,11,12,13,14,15,16,17,18,19,20,21,22]. In addition, 5G Technology would not only facilitate excellent communication services and accessibility to the Internet of Things but also be an excellent feasible solution for device-to-device connectivity. An automation in industrial manufacturing, such as tele surgery, automatic fracture detection application, remote sensing, telemedicine, and much more would be possible with advanced fashion and zeal. Due to heavy traffic and more interferences offered by different services, the Sub 1 GHz and the sub-6 GHz older spectrum bandwidths have become limited [11]. The term “bandwidth” refers to the most important property of an antenna system being responsible for the improvement of capacity to meet the aforementioned 5G design goals. Ultralow latency with higher data rates can easily be achieved and can be utilized, but we are still faced with the much higher ratio of challenges and limitations such as link losses, multipath effects, small-scale fading as well as short coverage [11]. To fully overcome and alleviate these challenges and limitations, different valuable solutions are available which might handle these issues, such as advanced beam-forming techniques, antennas with high gain and narrow beamwidth, small-cell technology in advanced format, massive MIMO antennas with excellent steerable characteristics of low scanning losses and reconfigurable characteristics [2,15,16], and bandwidth enhancement techniques [23,24,25,26,27,28].

For mitigating these challenges, antennas with high performance parameters such as antennas with small beam width, small side-lobe level, with characteristics of low scanning losses, are required, which might be able for two- and three-dimensional steering capability at higher levels of frequency scanning. It is a challenging and difficult task to maintain narrow beam width with low scanning levels. For flawless reliable communication, broadband antennas with excellent steerable characteristics, that are small in size, and have narrow beam width and high gain are required at mobile base stations.

The key challenges for designers is to maintain a suitable position of the antenna at base station and to maintain stability in the radiation characteristics with other communication components such as Wi Fi, 4G LTE, and LCD; besides these challenges, the back casing composed of metal is an enormous challenge for designers and researchers.

Specific absorption losses in terms of the effect of the user’s body on antenna performance are required to be inspected. Between each two antenna elements, the mutual coupling should be reduced or isolation between each two elements should be high [29], especially in the case of MIMO antennas. Some other performance parameters such as channel capacity loss (CCL), diversity gain (DG), mean effective gain (MG), and total active reflection coefficient also need to be analyzed [29,30].

The microstrip antennas are mostly used due to being lower in cost, simple in fabrication, and easy in installation, but they have some limitations such as limited level of gain, narrow in bandwidth, lower efficiency, and low power-handling capability. Various strategies and techniques are used to enhance the bandwidth, such as to create the slots in the radiator of microstrip patch antennas in the ground plane of antennas, as well as impedance matching techniques, increasing the height, and stacking arrangements of patch [2,3,15,16,17,18].

For microstrip patch antennas, these are conventional performance enhancement techniques in terms of bandwidth, but they effect some additional performance parameters such as gain, radiation pattern, side-lobe level, and cross-polarization level [25,31].

Simultaneous use of the coalescence of numerous technologies is needed in the designing of an antenna for enhancement of bandwidth, gain, and efficiency as well as reduction in side-lobe level and achieving a high level of isolation among interelements of a MIMO or array antenna. The above-mentioned techniques are famous conventional techniques for enhancement of performance, especially in bandwidth

Microstrip antennas are known for their narrow bandwidth, low gain, and low efficiency [31,32]. By using new technology, researchers are able to increase bandwidth by 70% and gain 30–35 dBi and various structural alterations. There are several methods for increasing bandwidth which have been employed by researchers in the evolution and development of 5G antenna systems for a flawless communication system.

The main focus is on the latest key techniques responsible for the improvement in bandwidth, gain, and efficiency, and in the case of MIMO, isolation enhancement is the main target studied in the literature [33].

First, we discuss in short the global fequency spectrum used by different regions for 5G application [32], as shown in Figure 1. The global snapshot elaborates the frequency ranges for each region where work for 5G communication is in progress. Under the umbrella of (ITU-R), the master international frequency register (MIFR) is the key element of international frequency management, which is a permanent database that contains the spectrum characteristics for radio stations’ operations throughout the world. The international recognition and protection against interference is conferred by these stations. This database is managed by BR and currently contains 2.6 million frequency assignments for terrestrial services, and over 200,000 are added every year.

More than 1.1 million assigned frequencies are contained in this database. In addition, about 350,000 assigned frequencies for the broadcasting-satellite service and 25,000 allotted frequencies for the fixed-satellite service are planned for future uses.

In [32], the authors discussed the global frequency spectrum as shown in Figure 1. A detailed summary of frequency bands both licensed and unlicensed are shown. Furthermore, the paper is split out into six sections. The first section of “Introduction” covers 5G technology’s goals as well as a summarized package of the literature review consisting of the latest state-of-the-art work focused on the latest performance improvement strategies and techniques in performance parameters. There are three main categories of frequencies in the assigned global frequency spectrum. The first one is assigned as high bands above 24 GHz, the midband frequencies range is from 1 GHz to 7 GHz, and the low band is below 1 GHz. Furthermore, these bands are divided into licensed bands, over 40 bands globally for LTE, which remains the industry’s top priority. the unlicensed spectrum is 2.4 GHz/5.9–7.1 GHz globwise.

After introduction, the second section of the paper is focused on 5G antenna trends for base stations or access points and for mobile terminals.

The third section of the paper summarizes the latest performance enhancement techniques discussed in detail in the literature review.

The fourth section is about the beam steer ability and beam forming techniques.

In section five, the reconfigurability for different environmental scenarios are discussed.

The sixth section presents a deep focus on the MIMO antenna performance-enhancement techniques such as mutual coupling reduction/decoupling techniques, which might effectively play a vital role in mutual coupling reduction.

The seventh section is the future challenges and opportunities.

Section eight is the conclusion, which concludes the paper in a summarized package.

There are a variety of strategies that must be used in conjunction with traditional antenna design techniques, data rate, signal-to-noise ratio and a lot more.

## 2. 5G Antenna Development Trends for Base Station (Access Point) and Mobile Terminal

The rapid exponential growth of bandwidth-hungry applications of current smart phone users have evoked researchers across the globe to redesign and update the existing commercial communication system to fulfill the near-future high data rate and wide bandwidth applications. The sub-6 GHz mmWave spectrum being overcrowded, researchers and engineers have focused to proceed toward efficient designing technologies and techniques to achieve higher bands in mmWaves instead of sub-6 GHz to overcome these issues [1,2,3]. For the near-future 5G communication systems, the feasibility of mmWaves being used as a carrier frequency is coming closer to the reality. The issue at the primary level is the inherent high path losses at mmWave frequencies, which need to be mitigated by high-gain antennas used on both sides: at the mobile terminal and at the base stations or access points. Before targeting the antenna structure for 5G applications for a base station, we focus on the basic scenario of 5G communication in a coexisting multiband scenario of fixed-earth stations and 5G base stations which would help us in selecting the techniques to be implemented in the designing of antennas for base stations or access points and mobile terminals. Base-station antenna technology (BSAs) and evolution details are as follows.

### 2.1. Evolution of (BSAs) Technology and Capacity Enhancement Techniques

First-generation networks are based on omnidirectional cells focused mainly on coverage and not on capacity. With each exponential increase in the number of users, the operators of the second-generation network started to search for more effective ways to overcome the main issue of capacity [4]. The main technique used for capacity improvement was sectorization. In 2G, the capacity-enhancement techniques used was to divide the existing omnidirectional cells into three sectors with a coverage of 1200 each. Each antenna has the capability of a 10 dB beam width of 1200 each. Polarization diversity in this era was a second techniques for capacity improvement at +450, with the rapid increase in the number of users moving from 2G to 3G as the mobile data services were introduced in 3G. A key factor for an effective increase in the capacity was to further subdivide the existing sectors into narrower sectors [5]. Half-power beam widths such as 65° or even 33° are shown in the Figure 2.

The main drawback was the increasing number of the antenna elements leading to overload in the tower. One solution was initiated to avoid the overloading of antennas by introducing the multibeam panel antennas techniques. Hybrid couplers were used for achieving multibeam capabilities, but they need comprehensive planning for managing the network to reduce cell-edge interferences, and operators need to adjust cell/sector coverage.

The use of phase shifters played a key role in network optimization. Two bands of 800 MHz and 900 MHz were introduced in this scheme, and 4G LTE with additional spectrum was introduced with up to 2.6 GHz.

### 2.2. 5G Antennas for Base-Station Communication Scenario

A conceptual scenario based on a practical cellular communication system is presented with a clear aim of understanding the transmission and reception of wireless communication signals at 900, 1800, and 2700 compared with a 450 beam shutoff range, and as a result, minor improvement in the exclusion zone is achieved [6]; as the beam shutoff range increases to 900, 1800, and 2700, the restriction zone then decreases significantly. Both zones can be achieved by loading the 5G BSs and ES actual parameters.

In this study, an angular protection scheme for fixed-earth stations (FEs) and 5G base stations is analyzed. Relative location, relative distance, as well as angular changes are analyzed. The angular protection scenario is the effective way to solve the coexistence-related issues of FSS and 5G BSs. The implementation of both the angular protection and distance protection should lead to more strengthening in the angular protection. The whole scenario is shown in Figure 3. The problem caused by interference can be solved effectively by a distance protection scheme. Four types of interferences are reported in [7] between FSS and 5G system. The types of interferences reported are as follows.

From the FSS earth station to 5G base station.

From the FSS Es to the 5G user.

From 5G BS to FSS devices.

From 5G users to FSS devices.

In the work previously discussed, we have focused on the calculation of interferences in the elevation, and azimuth planes are presented by the formulas reported in [2,16] as follows.
I_FSS ES→BS_ = P_BS tx_ + G_FSS ES_ + G_BS_ − PL
where I_FSS_ is the power density (dB/MHz) of the transmit station.

The gain of the antenna of the ES in the direction of the transmit station (in dBi) is represented by G_FSS ES_, where G_BS_ represents the gain of 5G BS by using MIMO antenna array in the direction of ES considering the antenna pattern(dBi) with beam-forming capability.

To properly compute the range of the beam shut-off angle ϕ_off_ (i.e., the size of the beam angle that must be shut off) the BS interference was calculated in each step. If the I/N interference threshold value remains less than that of the interference with the current beam sweep angle ϕs, then the beam angle must need to be shut off for protection of ES. The shut-off angle ϕ_off_ of the beam increases by 1°, as shown in Figure 4.

### 2.3. 5G Antenna Applications in Base Station or Access Point

Several researchers have explored antennas for base station or access points. For suppression of side-lobe levels, different techniques have been proposed in [1,2,3]. For phase error correction, the parasitic patches have been used with an excellent approach for aperture designing employed. Design with circular polarization is proposed in [9,10]. In [34], a standard dielectric substrate is used with integration of EBG unit cells periodically for suppression of the surface waves to achieve better radiation efficiency.

A planar array with high-density elementary radiators for next-generation 5G base stations is described in [35], yielding a compact size; the antenna is designed as densely interconnected in a stacked structure with high integral capability. According to the specific application-driven requirements, the optimization is performed in such a way throughout the entire planner arrangement to meet these requirements. Detail is shown in Figure 5.

At a resonance of 28 GHz, suitable performance in terms of realized gain, polarization purity, impedance matching, and inter element isolation is achieved. Out-of-band filtering capability and wide noncontiguous stop bands without any additional circuitry is obtained. The array is shown in Figure 5. Reference coordinates are shown in Figure 6.

### 2.4. 5G Antenna Applications in Mobile Terminal (Shared Aperture)

The realization of a shared-aperture sub-6 GHz and 5 GmmWave antenna system is proposed in [36] for application in mobile terminals or handheld devices. The antenna is designed in such a way that the integration for both bands of sub-6 GHz and mmWave operation is excellently achieved with the realization of both the simulated and measured results. The integrated antenna structure is composed up of a dipole and a tapered slot operating at multibands i.e., 3.6 GHz in sub-6 GHz and also the validation of 28 GHz in mmWave is performed. The tapered slot itself operates at 28 GHz and is also used to excite the dipole at 3.6 GHz. The simulated results of the tapered slot, rectangular slot, as well as the results for using the stub and without stub, are also shown in the figure. The dipole arms work as an antenna footprint. The designed structure has an overall size of 75 × 25 × 0.254 mm^3^ using Ro-5880 substrate. The validity is confirmed by matching the simulated and measured results. Mobile phone antenna designing is always the art of how to compromise between size, appearance, and performance. In recent years, metal casing for smart phones is gaining significant interest in the mobile industry. The main reason for using metal casing is their improved mechanical strength and attractive appearance and performance, as well as better thermal conductivity. The structure with results is shown in Figure 7.

### 2.5. Shared Aperture Using Reconfigurability Techniques

To design a common aperture for both mmWave and sub-6 GHz, the authors have used reconfigurability techniques for 5G smart phone applications. In [11], a microstrip patch through a PIN diode is linked up with a meanderline structure to achieve the reconfigurability between the two desired frequency bands. An excellent MIMO performance is observed across the entire bandwidth for both proposed bands of mmWaves and sub-6 GHz to fulfill the requirements of an efficient MIMO antenna system. The antenna geometry and results are shown in Figure 8.

### 2.6. Shared Aperture Using Reconfigurability Techniques

Integration of sub-6 GHz and mmWave band is a challenging task to maintain the compactness of 5G smart phones owing to large frequency ratios. In [12], a microstrip patch radiator loaded with an inverted u-shaped slot to ensure dual-band operation in the mmWave and sub-6-GHz frequencies, i.e., 2.8 GHz and 38 GHz. In the proposed design, a patch radiator is linked up with a meander radiating structure through a compact microstrip resonant cell (CMRC) as a low-pass filter and a 12.9%, 5.8%, and 2.4% wide decoupled bandwidth are achieved.

## 3. Antenna Performance Enhancement Techniques

Antennas are one of the pivotal parts of 5G devices and a main part of a communication system. Due to the natural phenomena of using microstrip technology, especially in 5G, plenty of radiation losses as well as issues in the key performance parameters are presented, such as narrow bandwidth, lower efficiency, and less gain. To overcome all these burning issues, we need to design such antennas to fulfill the very basic requirements in terms of wide or broad bandwidth, high gain, and high efficiency, which would provide an excellent provision for low latency, ultrareliable flawless communication with better spectral efficiency. The one and only solution is to exercise such excellent techniques which might be able to overcome these issues and make the design as the best solution. The main focus is on the latest key techniques responsible for improvement in bandwidth, gain, and efficiency, and in case of MIMO isolation enhancement is the main target discussed in the literature. First, we elaborate the layout of all key performance parameters to summarize our study and make a provision for better understanding other researchers’ working in the same field. Researchers have also worked to find out the human body effects on the performance of antenna functionality. It is a challenging task for research industry to practically implement the techniques to minimize these effects and make functionality and efficiency of antennas better. Some other challenges, such as the effects of high-frequency signals on the human tissues are also key challenges to be addressed.

### 3.1. Effect of Substrate Choice on Performance Enhancement

The appropriate selection of a substrate is the main requirement of an antenna implementation for valuable and fruitful results, as in [37]. For antenna fabrication, various substrates with different permittivity and loss tangents are available to be used based on choice and requirements of a researcher. For enhancement of gain and reduction in power loss, the selection of a substrate could play a vital role. Substrates with less relative permittivity and low loss tangents must be selected to achieve the goal of enhanced value of gain and a decreased value of power loss.

### 3.2. Effect of Corrugation on Bandwidth and Front-to-Back Ratio

Removing the metal part, which might be in any geometrical shape such as square, rectangular, triangular, sine or any other), from the radiator’s edge has a great impact on improving the most important performance parameters in terms of bandwidth, gain, and front-to-back ratio [38].

### 3.3. Impact of Dielectric Lens on Directivity and Gain

Electrostatic radiation transmitted in one direction by the dielectric lens has the capability to divert the direction of the electrostatic radiation in one direction, which leads to an increase in the directivity as well as gain of an antenna, as in [39]. Different geometrical shapes of dielectric lens are available, and they are designed by alignment with the same or different substrate materials.

### 3.4. Multielements

The gain of an antenna can further be increased by using the antenna with multielement. It also improves the bandwidth and efficiency of an antenna, as in [40]. In applications where a single element antenna cannot fulfill the high gain and wide bandwidth requirements, a multielements antenna is the best solution and is more effective in this case.

### 3.5. Effect of Dielectric Resonator on Gain and Bandwidth Enhancement

How to achieve a high gain and wide bandwidth a dielectric resonator antenna (DRA) for 5G wireless communication is proposed in [18]. In a sequence, to achieve a high value of gain, the DRA antenna is operational at higher order mode TEx δ_15_, while to improve the bandwidth, the quality factor is reduced by employing a hollow cylinder at the center of the DRA, as shown in Figure 9. A 50 Ω micro strip line with a narrow aperture slot was used for excitation of DRA. For the TEx δmm mode, the resonance frequencies can be extracted from the dielectric waveguide model (DWM); the three waves K_X_, K_Y_, and Kz could be extracted by solving the transcendental equation as follows.
(1)$$k_{x}\tan\left(\frac{k_{x}a}{2}\right)=\sqrt{\left(\varepsilon_{r}-1\right)k_{0}^{2}-k_{x}^{2}}$$
(2)$$k_{x}^{2}+k_{y}^{2}+k_{z}^{2}=\varepsilon_{r}k_{0}^{2}\;$$
(3)$$k_{0}=\frac{2{\pi f}_{0}}{a},k_{y}=\frac{m\pi }{b},k_{z}=\frac{n\pi }{d}\;$$
(4)$$f_{0}=\frac{c}{2\pi \sqrt{\varepsilon_{r}}}\sqrt{k_{x}^{2}+k_{y}^{2}+k_{z}^{2}}$$

### 3.6. Substrate-Integrated Waveguide Feeding Techniques (SIW)

Substrate-integrated waveguide (SIW) techniques have so many excellent features, on the basis of which one can easily chose such a cheaper and effective way, especially in the designing structure of antenna for 5G application. In such techniques, the radiation losses are comparatively lower as in the conventional feeding mechanism. There is much possibility of radiation losses, especially in 5G antennas, so the power manipulating capability is higher than the one in conventional feeding ways. Various RF components which use SIW techniques are much cheaper in cost. In the process of mounting, the discrete components on SIW structures have a high density of integration.

A substrate-integrated waveguide broadband, with high gain and low-cost features, is presented in [19]. For 60 GHz bands, a single-layered SIW feeding network for achieving high gain with wide bandwidth is employed with a combined structure of cavity-backed patch antenna and wide-band T-junction. Although the antenna has a multilayered structure, it can still be easily printed with single-layered PCB techniques. There is a unidirectional symmetrical radiation pattern with higher gain of 19.6 dBi with lower cross-polarization characteristics securing 27.5% impedance bandwidth. In [20], a substrate-integrated waveguide feeding structure is used for wide-band characteristics, the SIW antenna offers a wideband (25.2–30.2) GHz with a peak value of stable gain 16.4 dBi for orthogonal polarization discrimination above 30 dB with high efficiency above 30 dB.

To suppress the cross Polarization in the single Element a coupler with four way broad wall for 2*2 subarray is applied a 4*4 main Array exhibiting 26.7% Bandwidth is designed on the basis of two sub Array having a peak Gain value of 26.7 dB are shown in Figure 9. A substrate-integrated waveguide SIW-fed array antenna for 5G application with broadband and high gain characteristics is demonstrated in [21]. A two-layer SIW feeding network of the aperture coupling method is adopted. For the purpose of broadening the bandwidth, the heights of the posts are designed as lower than the height of the cavity. Moreover, for in-band duplex communication scenarios, this SIW technique is employed. The peak self-interference cancellation value of >36 dBi is achieved. The full duplex communication scenario is shown in Figure 8. SIW provide the benefits of planner integration. The proposed design integrates a dual linear polarized three port differential antenna, a three-port SIW common-mode power combiner, and a 180° phase shifter at 28 GHz. The system operation is shown in Figure 10.

A stacked microstrip patch antenna array with SIW feeding structure is used for wideband application. In [22], an antenna element with linear polarization and wide bandwidth of 83.75 from (50–74) GHz with 8.7 dBi value of average gain is demonstrated. In [22,37,38,41,42,43,44], with different strategies, the SIW techniques have been used and effective results in terms of gain, bandwidth, and efficiency were achieved. Advantages and disadvantages of different techniques are shown in Table 1.

### 3.7. Effect of Slots in Radiator with Different Geometrical Shapes

In [31], a patch antenna in a rectangular shape with a triangular slot loaded on the surface of the radiator on the right side from top to bottom is presented for wide-band applications. The combined effect of the triangular slot and partial ground plane excellently increases the impedance bandwidth more than 90% efficiency and 12 dBi gain is achieved. The impact on bandwidth is shown in Figure 11.

An ultra-wideband antenna with a truncated “U”-shaped slot in the patch is presented in [32]. A compact size 2.9–23.5 GHz and a LTY-5 Taconic εr = 2.2 substrate is used; the lower corner of the patch is modified with a semicircle. For the wide-band characteristics’ realization, a slot in U shape is etched at the center of the radiator to ensure achievement of the notch band. A Structure in the semi ring shape in [26] has excellent impact on bandwidth enhancement.

In [24], for enhancement of impedance bandwidth, leaky wave half width microstrip patch antenna is presented. The enhancement of bandwidth is realized by etching four circular slots on the surface of the radiating patch. The main beam with a wide scanning range of +12 degrees and +70 degrees is achieved where range frequency sweep is 4.28 GHz to 7.13 GHz with peak gain of 10.31 dBi at 5 GHz. A high radiation efficiency with main-beam continuous scanning capabilities in forward direction only by changing the operating frequency is achieved. Due to the circular slots in the radiation element, the level of cross-polarization decreases.

In [57], a multimode characteristic E-shaped patch antenna for 5G mm wave application is presented. A slot is introduced in the proposed design and a 45.4% wide impedance with low loss polarization and stable radiation pattern is achieved.

### 3.8. Role of Parasitic Patches in the Band and Gain Enhancement

The use of parasitic patches has shown an outstanding role in the enhancement of performance parameters, with especially effective behavior in the bandwidth enhancement. In [25], the authors presented a simple low-profile spiral monopole antenna upgraded with vias in the ground plane in the inverted-“U”-shaped sequence. A 11.5 dBi peak gain with more than 83% radiation efficiency is achieved. The back side of the antenna is introduced with parasitic patches in the hexagonal shape to restrain the flow of surface waves and to minimize the interelement mutual coupling. Bandwidth ranging from 23.76 to 42.15 GHz is achieved. In the geometry of the antenna, two spiral arms are lengths L1 and L2, respectively. These arms are then connected to the feed line. A rectangular slot is created in the back side of the antenna. The width of the slot is the same as in the feed line. RT/Duriod Rogers 5880 is used with dielectric constant εr = 2.2, and the thickness of substrate is 0.254 mm. A low-profile parasitic antenna in G shape for 5G is presented in [27]. The antenna offers the impedance bandwidth with two separate ranges. For the result-oriented geometry of the parasitic design, the antenna is optimized and a wide bandwidth due to the parasitic patch was observed, as in Step 1, and as 7.46 to 8.42 GHz in the second step from 4.43 to 8.82 GHz.

The authors of [27] have cross-verified the results by cross-checking on both software CST microwave studio and high-frequency structured simulator HFSS results from both software are well agreed. For converting the end fire direction and pattern of beam to the broadside, the extension in the U-shape have a key role.

Similarly, in [28] the parasitic elements are placed on either side of the monopole radiator to improve the bore-sight gain. The mutual coupling between of two adjacent elements is reduced by the arrangement of radiating patches and their feed lines with a rotation of 180 degrees out-of-phase alternating sequence discouraging the flow of surface waves, which directly results in improving the symmetry of the radiation pattern, and mutual coupling of 20 dB with uniform distribution of amplitude and a 19.8 dBi maximum broadside gain is achieved with a side-lobe level of −12.1 dB. These characteristics rendering the antenna would be an excellent choice for 5G wireless applications. The geometry of the parasitic patch is shown in Figure 12.

The coupling of 20 dB with uniform distribution of amplitude and a 2 dBi maximum broadside gain is achieved and a side-lobe level of −12.1 dB is achieved. All simulated parameters well agree with the measured one.

### 3.9. Role of Hybrid Feeding Structure in Performance Enhancement

A hybrid-feed wide-band and high-efficiency antenna for mmWave 5G communication is proposed in [58]. The proposed designed antenna consists of a novel hybrid-feed structure in a planar antenna array with excellent performance in terms of high efficiency and wide bandwidth is presented. To achieve the goal of high-performance key parameters such as wide bandwidth and high efficiency, the conventional high-loss ridged feeding network and substrate-based feeding network are replaced by an effective low-loss ridge-gap waveguide feeding network. The measured bandwidth ranges from 26.05 to 31.15 GHz (16%), and the maximum achieved gain is 25.15 dBi. The proposed design might be an excellent choice for 5G application. In comparison with [38], the results are much improved due to the use of ridge gap waveguide (RGWG) instead of gap waveguide GWG. Structrual detail with different layers of the design is shown in Figure 13.

### 3.10. Differential Feeding Network (Specific Feeding) 

In [39], a patch antenna fed differentially with wide-band and dual polarization capability with excellent filtering response is proposed. For excitation of square patch antenna, a feeding structure with novel E-shaped is proposed. One end of the feeding network is shortened to the ground for achieving the wider operation bandwidth, also an extra resonant mode is introduced for generation of a third resonance mode. A stepped.

Resonator with symmetric short/open-circuit is proposed to further enhance the bandwidth. The same resonator not only makes feasible the enhancement in bandwidth but effectively enhances the gain suppression level outside the operation band. For an active VSWR value of fewer than 1.5, a wide operation bandwidth (3.12–3.9 GHz at 22.2%) is achieved with in-hand stable gain suppression level for out-band is better than 18.5 dB. The geometry of the differential feeding network is shown in Figure 7 as follows. The radiating patch and E-shaped feeding structure are printed on the top layer of the upper substrate. One end of each E-shaped feeding structure is shorted to the ground with metal posts. The lower layer is a F4B substrate with a dielectric permittivity of 2.65 and thickness of 0.8 mm. In addition, four pairs of O/SCSIRs are printed on the top layer of the lower substrate. The four ends that close to the center of the substrate are connected to the E-shaped feeding structures with metal posts. The other four ends of the O/SCSIRs are connected with 50 Ohm coaxial cables. A metal ground is printed at the bottom layer of the lower substrate with four U-shaped slots. When port1 and port2 are excited with differential signals, a 45° polarization is generated. Similarly, a 45° polarization can be obtained when port3 and port4 are excited with differential signals. When the shorting posts are introduced (the parallel inductance L2 is added), the overall inductance decreases, and the first resonant mode moves to lower frequencies. Thus, the two resonant modes are separated by the shorting posts, and a wide operation band is obtained. The proposed structure with metal post and upper substrate is shown in Figure 14.

### 3.11. Performance Enhancement Using Metamaterials

Artificial materials are composed up of periodic material structures and specific dielectric whose electromagnetic properties in terms of electric permittivity as well as magnetic permeability could be redesign to a level higher than the nature of the available materials. Materials having negative permittivity and negative permeability do not exist in nature, and metamaterial is the one and only source. If we replace conventional patch antenna with MMTs, which are commonly used with DNG double-negative characteristics, it will lead to an excellent shift to higher frequencies, and thus will definitely increase bandwidth [59,60]. In [61], a modified patch with diagonal slot and a layer of metasurface is demonstrated in low profile with circular polarization. A patch with a diagonally positioned slot in the center of the patch to achieve wide bandwidth as well as circular polarization is proposed. The metasurface is designed without generating any gap on the patch surface to make provision for miniaturization. A wide-band low-profile antenna with circularly polarized metasurface-based antenna for realization of performance enhancement in terms of high gain, wide, bandwidth, high efficiency, and circular polarization is presented, as well as a wide bandwidth of 37.4% (24–34.1 GHZ) for the value of S11 < −10 dB with a high value of gain 11 dBic. The gain response is stable at 9.5–11 dBic. CP in the left hand is achieved. At 11 dBic the same Antenna is tuned. Large axial ratio as well as high bandwidth and high efficiency is achieved. Similarly, in the antenna in [62], a circularly polarized structure with wide band and bidirectional metasurface-based antenna is proposed. A metasurface composed up of two identical periodic metallic layers as a metal surface is used. In this case, the metasurfaces convert the linear polarized radiation to circular polarized radiation which was produced by the rectangular slot. In total, 14.3% bandwidth is achieved (5.2–6.0 GHz). The antenna offers state-of-the-art operating bandwidths compare with simple planar antennas, shown in Figure 15.

## 4. Beam Steerability and Beam Forming

### 4.1. Dielectric Resonator with Beemsteerability

A microstrip patch antenna is presented in this research work. The front and back side of the dielectric slab is arranged periodically to properly adjust the VCRM position. A good radiation pattern is attained with good matching of numerical and experimental results. A high value of 11.9 dB gain is achieved in [62]. A dielectric resonator antenna with parasitic elements is used to achieve high-order mode steerability. The beam steering was successfully achieved by switching the termination capacitor on the parasitic element and a narrow aperture in the ground plane to achieve a wider bandwidth, which can be potentially applied for device–device (D2D) communication in 5G Internet of Things (IOTs), which can interconnect, as shown in Figure 16. For a significant performance achievement, the size of the resonator might be less than the device antenna [63], and 36.78% impedance bandwidth is achieved. The entire design mechanism is shown in Figure 16.

A 193.16% in [57,64] is achieved and four CSRRs in the ground plane are used. A total of 5.71% impedance bandwidth while the simple patch Antenna without the use of CSRRs offer an impedance bandwidth of 4.28% at 28 GHz.

### 4.2. Impact of Fabrication Techniques

Antennas need to be robust mechanically and should be efficient with wide bandwidth as well as having desirable radiation characteristics [65,66,67,68,69,70,71,72] which might ensure high simulation accuracy [31]. Radiation pattern and gain of the antenna are required to be tested for distortion or degradation. In the light of the literature studies, here is the summarized comparison on the bases of advantages and disadvantages in terms of bandwidth, efficiency, and gain. A brief summery has been studied in the literature about how different fabrication techniques affect the performance of an antenna. A summary of different antenna performances along with fabrication techniques is presented in Table 2.

## 5. Reconfigurability

Since past decades, reconfigurable antennas have gained considerable attention [33,73,74]. Unlike wideband and multiband antennas, the reconfigurable antennas have excellent feasibility for customers to select a frequency band on their own [74]. For facing different environments with different complexities, the response of the antenna is a key asset for flawless, smooth, and reliable communication. For several applications such as searching, rescue, and tracking, the antenna needs to have effective switching capabilities as well as performance-boosting technologies. To handle different applications scenarios, different switching techniques, phased arrays with adaptive nulling characteristics, and multiple beams with low side lobe are required. For achieving such multiple tasks, different kinds of signal processing techniques can excellently provide effective solutions. A high degree of reconfigurable antenna is required for reliable communication. In [33], a low-profile simple reconfigurable antenna is proposed. Two pin diodes are used, as shown in Figure 17. These pin diodes connect the two additional stubs and the triangular monopole. The same techniques are used in Figure 18.

The antenna has a feasibility in two different bands and when the two switching states are ON, thus the antenna operates in a single band of 3.3 to 4.2 GHz. In the case of only one switch being in the ON position, the antenna then offers the dual band ranging from 3.3 to 4.2 and 5.8 to 7.2. The antenna with low profile and compact size of 0.27λ0 × 0.16λ0 × 0.017λ0 can be used for several applications, such as for wireless systems in vehicular systems and wireless local area networks (WLAN). A bidirectional beam characteristic is achieved. 

In [33], in the first stage, the simple patch antennas is modified to a right-angle triangle and then two stubs are introduced on the top angle of the triangle in an inverted L-shape for enhancement of bandwidth. Two diodes, D1 and D2, were placed between the two stubs and the radiator. The antenna offers the broadside radiation pattern when both the diodes D1 and D2 are in the OFF state in H-plane while in the E-plane a small tilt is observed in case D1 is OFF and D2 is ON. On the other hand, when D1 is ON and D2 is OFF, it provides pattern radiation in 30 GHz and 32 GHz, respectively. When both diodes (D1 and D2) are in the ON state, then a little tilted beam is achieved by the antenna. The tilt scenario is due to the truncated ground. When both diodes are in the ON state, the impedance bandwidth offered by the antenna is 6.3 GHz, ranging from 29.95 to 32.25, which covers the potential spectrum from 26–29 GHz for 5G applications. When both diodes are in the OFF state, there is a bandwidth of 6.75 GHz ranging from 27.81–34.560 GHz, where S11 < 10 dB. In case (01), when one diode D1 is ON and the second one is OFF, the impedance bandwidth is of 6.42 GHz (27.81–42 GHz), where S11 is < −10 dB.

The design is shown in Figure 19.

After generalizing the view of the reconfigurability scenario, the frequency and pattern reconfigurability must be explored. In [27], lumped elements as a switching component for hexaband switching capability is presented. Lumped elements are used to achieve tunable capacitance responsible for frequency reconfigurability of hexaband frequency spectrum, i.e., 2.10 GHz, 4.11 GHz, and 2.4 GHz and 5.2 GHz on affordable, compact geometry capable to be integrated with ease in modern communication systems.

Pattern reconfigurable antenna is presented in [72]. For 5G, there are new radio frequency bands –N77(3.3 GHz–3.8 GHz). The structure of the radiating element is modified by a swastika-shaped structure. Around the radiator, four parasitic arc-shaped elements are loaded and each parasitic element is loaded with a pin diode to control the antenna radiation field pattern. Table 3 is the comparison of different antennas using pin diodes for switching.

A UWB antenna [80] with improved gain and omega structure loaded in the radiating patch is presented. In omega structure, a lumped capacitor is inserted for tuning the notched frequency spectrum 5.7 GHZ to 3.8 GHz for both WLAN and WIMAX bands, respectively. For gain improvement, a 3 × 2 single layer array with a unit cell of metamaterial in L-shape is used for UWB and gain improvement. Lumped capacitors have an effective role for the mitigation of interference of WiMAX and WLAN spectrum. Antennas with reconfigurable techniques through electronic switching components such as PIN diodes, Varacter diodes, and RF MEMS are reported in [33,73,74,81,82]. Each technique has some supremacy or advantages as well as some weaknesses/disadvantages. A comprehensive study on many kinds of electronics switching components used for gaining reconfigurability are also reported in [33,73,74]. Photoconductive switches are composed up of the famous semiconductor materials silicon and gallium arsenide [29,30]. Biasing lines are used instead of electrical wire, obviating large sizes and offering high isolation along with fewer interference. The details of different switching techniques are summarized in Table 4.

## 6. MIMO Antenna Performance Enhancement Techniques

In recent years in communication systems, the topic of multiinput and multioutput (MIMO) systems has received a considerable attraction, especially the performance in terms of the channel capacity of a communication system which can badly be affected by the mutual coupling of closely correlated interelements of a MIMO system. The capacity gain of a communication system has a significantly deep impact on the overall efficiency of an antenna system. So, a good communication system needs to have good isolation between the interelement or closely collocated elements.

A summary of MIMO antenna designing is reported in [81]. Most authors have used one or more techniques for designing antennas for mutual coupling reduction in MIMO antennas described in the literature for 5G and wide-band applications, these designing and decoupling techniques can effectively be used and better results could be achieved. The detailed layout is shown in Figure 19. The most prominent techniques of MIMO antenna designing as well as mutual coupling reduction techniques are presented to better explain the idea of mutual reduction or decoupling methods and possibilities discussed in the literature from [29,30,73,74,81,82]. In multielement antennas, either in arrays or especially in the case of MIMO antenna design, the collocated antenna elements on a single PCB might affect the performance of each other. To avoid or reduce this effect, researchers have implemented different strategies and techniques for mutual coupling reduction or decupling of these collocated elements n the case of MIMO antennas, which is the main focus in our study, named as isolation or decoupling strategies and techniques in the literature. Some state-of-the-art techniques are explained in the next section. The decoupling techniques play an excellent role to achieve the optimum performance in case of MIMO antennas. These techniques are a predetermined and unavoidable part in the designing procedure of the MIMO antenna. These techniques are explained below.

### 6.1. Mutual Coupling Reduction Using CDRA in Ground

A CDRA-based antenna is presented in [81,82]. In [82] the CDRA is used back to back on the opposite side of the substrate. For excitation of Port no.1 and Port no.2, a pair of coplanar waveguides is used with conformal microstrip lines. CDRA is placed on yop with a height of 6.5 mm and on the bottom with 6.0 mm. The excitation of Port no.3 and port no. 4 is conducted by microstrip line conformal fed to stripline. An improved interelement isolation is realized by generating two modes HEx and in each CDRA. Two different polarizations are produced due to the two different modes HEx and HEy. Excitation from opposite directions and two orthogonal modes in opposite directions make it feasible to excellently improve the isolation of closely colocated elements. DRA have shown better results in terms of bandwidth and gain [82]. In [30,62], monopole antenna with inverted “L”-shaped geometry and its MIMO design with eight elements is demonstrated. Good efforts have been carried out in [29]. In [30] a MIMO antenna the size of 136 × 68 mm^2^ with eight elements covering 450 MHz bandwidth is presented.

Isolation with value of 15 dB as well as high Gain of 4 dBi is achieved. In the structure of the antenna, a parasitic patch is extended from the ground plane. Excellent impedance matching in the desired frequency spectrum. MIMO antenna designing and mutual coupling reduction or decoupling techniques are shown in Figure 20.

### 6.2. Neutralization Lines

Having capability to pass electromagnetic waves between antenna elements to completely decouple or reduce the effect of mutual coupling by using lumped elements or metallic slits. It improves the antenna bandwidth and reduces the occupying space of antennas. The improvement in bandwidth is realized

### 6.3. Decoupling Network

Cross admittance becomes transformed to purely imaginary value in decoupling network by adding a discrete component or transmission lines. To reduce mutual coupling, this technique employs a plane decoupling network which acts like a resonator. The pattern diversity for multielements could also be achieved through decoupling by implementation of dummy load and coupled resonators technique. To improve isolation, it is a cost effective way to be implemented [88].

### 6.4. Electromagnetic Bandgap (EBG) Structure

For the transmission of electromagnetic waves, it acts like a medium. EBG structure provides capability of low mutual coupling and excellent efficiency, as in [88]. An improved scattering parameter with exhibiting wideband gap characteristics is achieved. In the desired direction far field gain pattern is achieved.

### 6.5. Dielectric Resonator

An antenna that contains a dielectric resonator is called as dielectric resonator antenna (DRA). DRAs are capable of providing low loss, high gain, and high radiation efficiency. Dual-band property as well as high isolation value can be achieved using DRAs [88].

### 6.6. Defected Ground Structure (DGS)

Different geometrical shapes, defects or slots are consolidated in the ground plane of the antenna. Low mutual coupling and wide bandwidth with maximum efficiency is the structure with defected ground in [88].

### 6.7. Metamaterial

Metamaterials are materials which can be designed manually by using two or more than two different materials which contain electromagnetic characteristics. Different types of metamaterials with the characteristics of single negative, electromagnetic, electromagnetic band gap, double negative, anisotropic, isotropic, terahertz, chiral, tunable, photonic, frequency selective surface-based, nonlinear, and tunable metamaterial. Using metamaterial, it is possible to have an antenna with low mutual coupling, high gain, bandwidth, and be compact in size [37,88]. Slot elements are used to achieve enhancement in the impedance bandwidth using the coupling method in the radiator or patch or in the ground plane. The slot antenna have the capability to provide wide bandwidth, high gain, and high efficiency but have a high mutual coupling value [89]. Complementary split-ring resonators (CSRR) are used for isolation improvement, to perform iterating function, and to provide lower mutual coupling. CSRRs are also used to provide high efficiency and miniaturize the size of the antenna [88]. CSRR is made up of two concentric ring structures with slots opposite to each other. Frequency reconfiguration is based on switching techniques by using varactor diodes and MEMS switches, and PIN diode reconfigurability can be achieved to increase frequency range and enhance the value of the envelop correlation coefficient.

### 6.8. Recnfigurability

The reconfigurable antenna structure can provide lower mutual coupling as well as a high value of diversity gain, and high efficiency, wide bandwidth, and front-to-back ratio can also be achieved [30,33,73,74,81,82,88]. These antennas are either a single element or multielement. The main focus of these enhancement techniques is to improve the antenna performance throughout in terms of mean effective gain (MEG), envelop correlation coefficient ECC, and 38.1 bps/Hz channel capacity.

A comprehensive and summarized detail of MIMO antenna designing techniques as well as performance enhancement techniques in terms of mutual coupling reduction is shown in Table 5. Table 6 elaborates the comparison of different techniques employed for mutual coupling reduction/decoupling. in this summery, the comparison of different techniques employed for mutual coupling reduction/decoupling is presented.

Some techniques are used not only for gain and bandwidth enhancement in single element and arrays but also used for mutual coupling reduction or decoupling the correlated closed elements’ place on the same single PCB. Like metamaterials, researchers have used metasuface structure for bandwidth enhancement in [31].

The same metameterial plays a vital role in MIMO as it is used for mutual coupling reduction due to its unique double-negative (DNG) characteristics. Different geometrical shapes are used by different researcher for achieving the goal of wide bandwidth, high gain and mutual coupling reduction. The role of metamaterials could not be ignored as they have excellent characteristics.

The Antenna in [82] offers 2.6 times a 4 × 4 MIMO antenna. All parameters effecting the performance of antenna such as SAR in terms of user hand effect is also taken in consideration.

## 7. Future Challenges and Opportunities

Novel circuit antenna co-integration needs to be a solution for active beamforming antenna arrays for 5G telecommunication applications in the sub 6 GHz (FR1) and near-30 GHz (FR2) mmWave frequency bands. Active integrated antennas can be designed with nontraditional functionalities (e.g., tailored power combining, active load modulation for power amplifiers [111], active source modulation for low-noise amplifiers, and RF filtering [112]) that help to minimize power loss, particularly at mmWave frequencies.

Antenna arrays with high directivity are implemented for fixed-beam communication at frequencies beyond 100 GHz. Most high-efficiency mmWave array antennas are waveguide (WG) structures that are typically designed with multilayer H-plane split blocks that are galvanically connected. This method results in a hollowed WG multilayer structure formed by stacking machined metal plates.

Due to crucial connectivity power dissipation and physical constraints, conventional systems incorporating phase shifter ICs attached to each element may not be practical at 100G+ frequency. Other systems, such as those based on reconfigurable antenna elements equipped by varactor or PIN diodes, suffer from comparable problems, as well as design complexity, high cost, and time-consuming assembly.

Innovative array antenna designs enabling two-dimensional (fullspace) beam control and effective beam steering are ideal for integration and hybrid packaging alongside active IC modules. Antenna subarray modularization (ASM) is one of the important technologies in this context for making the system practical and robust [112]. ASM will be investigated in order to determine optimally sparse topologies to relax the physical restrictions of antennas while meeting beamforming objectives with a lower number of amplitude-phase controls per element and remaining design compatible with semiconductor technology. One specific goal is to reduce signal latency while operating several phase shifters along the same ASM module.

The micromachined THz system platform also allows for the implementation of MEMS (micro-electromechanical systems) switches, allowing for reconfigurable antenna front ends. There is significant potential to realized silicon-micro-machined antenna systems with low-order (1 to 2 bit) MEMS-based phase control to support beamforming abilities of large-scale array antennas, with the benefits of low insertion loss, high compactness, convenience of control circuitry, and high-power packing abilities. The disadvantage associated with the limited number of degrees of freedom in beamforming can be overcomes by combining the above-mentioned array antenna topologies.

## 8. Conclusions

In communication systems, antenna plays a vital role and could be considered the important asset of a communication system. The main focus in this research is the study of state-of-the-art performance improvement strategies and techniques used for bandwidth, gain, and efficiency enhancement, as well as the mutual coupling reduction techniques used in the array of multiple elements or MIMO antenna elements. Different strategies and techniques used for excellent enhancement in the performance of antennas used for 5G and ultrawide band applications has been reviewed. Metamaterials play a multidimensional role in the performance enhancement of an antenna, especially for 5G and wideband application. MMTs not only play a role in the enhancement of bandwidth but have the capability to reduce mutual coupling between the closely collocated antenna elements in MIMOs and the arrays of multiple elements. The role of the EBG structure slots in the radiators with different geometrical shapes in the antenna performance enhancement in terms of bandwidth, gain, and efficiency is included. Reconfigurability is focused as having an important role and solution for different environmental complexities such as surveillance, tracking, and smart-cities scenarios. Reconfigurability techniques for sharing the same aperture for MMwave and sub-6 GHz application is also included. The role of beam steerability and beam forming is studied in detail.

## Figures and Tables

**Figure 1 micromachines-13-00717-f001:**
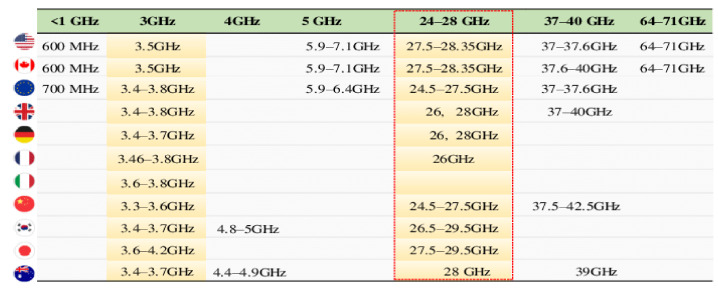
Snapshot of global frequency spectrum.

**Figure 2 micromachines-13-00717-f002:**
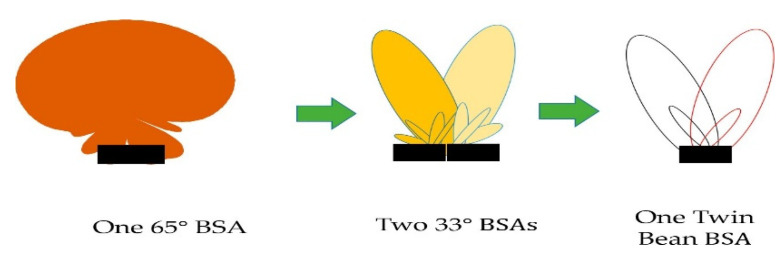
Single-beam BSA, two narrow-beam BSAs, and a twin-beam BSA [5].

**Figure 3 micromachines-13-00717-f003:**
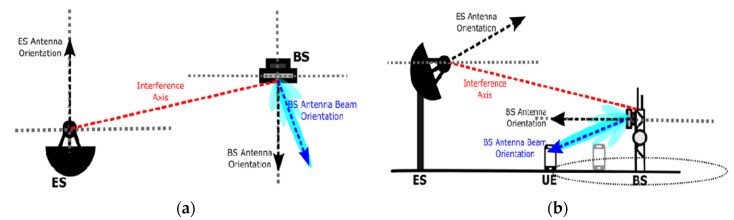
Carrier frequency set to the center frequency [7] in the (**a**) azimuthal and (**b**) elevation planes.

**Figure 4 micromachines-13-00717-f004:**
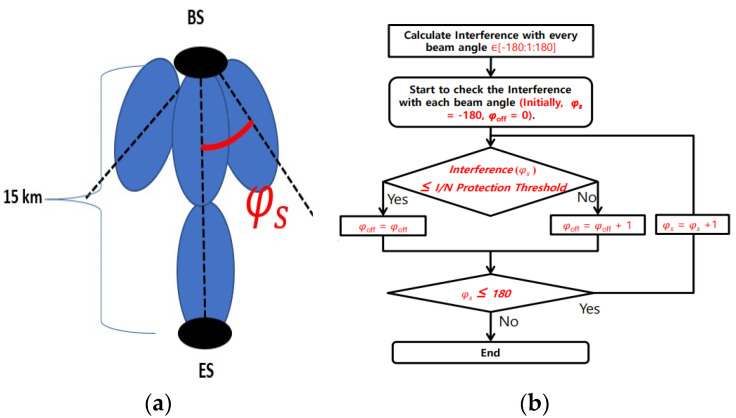
Fixed base station—earth station: (**a**) relative azimuth and (**b**) flux gram [7].

**Figure 5 micromachines-13-00717-f005:**
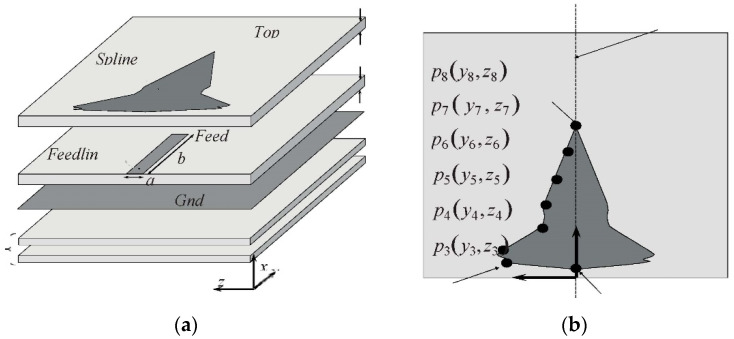
Geometry of the HDI—based filtering 5G patch radiator [35] (**a**) 3D and (**b**) top views.

**Figure 6 micromachines-13-00717-f006:**
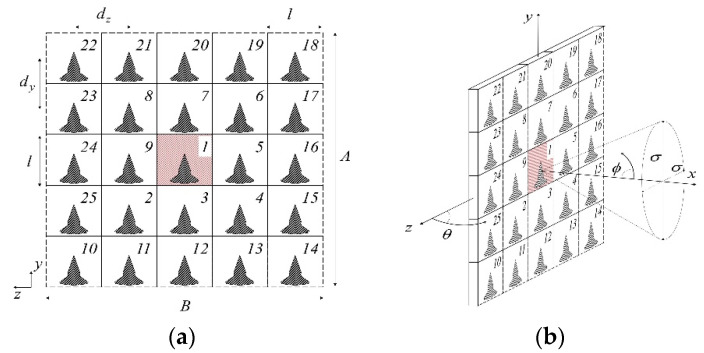
Planar array for 5G BSAs. (**a**) Front view and (**b**) the central embedded element identifying the referenced coordinate [35].

**Figure 7 micromachines-13-00717-f007:**
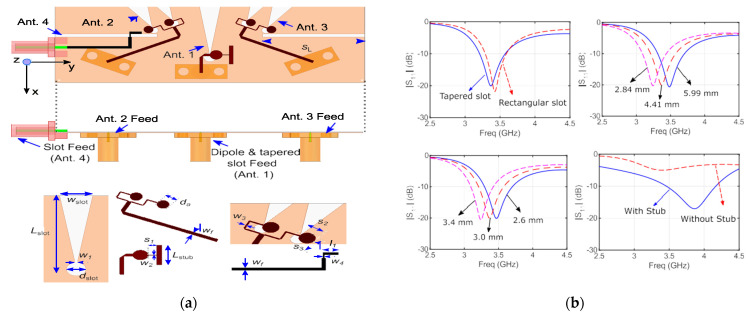
(**a**) The design geometry of sub-6 GHz and mm wave common aperture 5G antenna system [36] and (**b**) results.

**Figure 8 micromachines-13-00717-f008:**
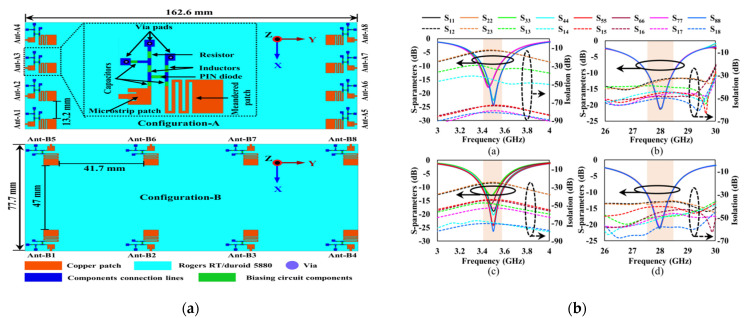
Antenna configurations (**a**) reconfigurable 8 × 8 MIMO system and (**b**) S11 parameters and isolation [11].

**Figure 9 micromachines-13-00717-f009:**
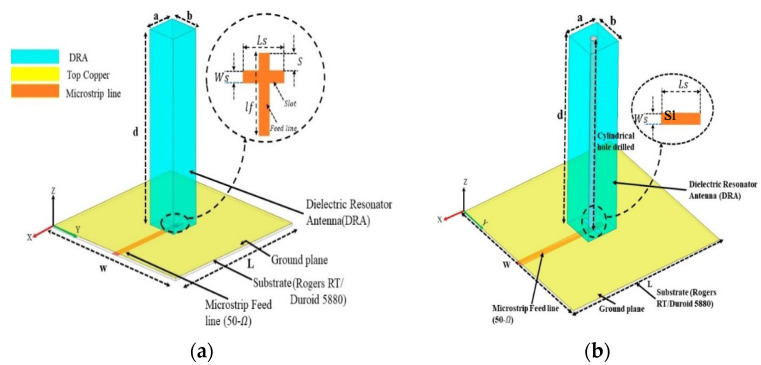
Configuration of (DRAs): (**a**) (DRA1) without hole TEδ15 mode; (**b**) (DRA2) with cylindrical hole TEδ15 mode [18].

**Figure 10 micromachines-13-00717-f010:**
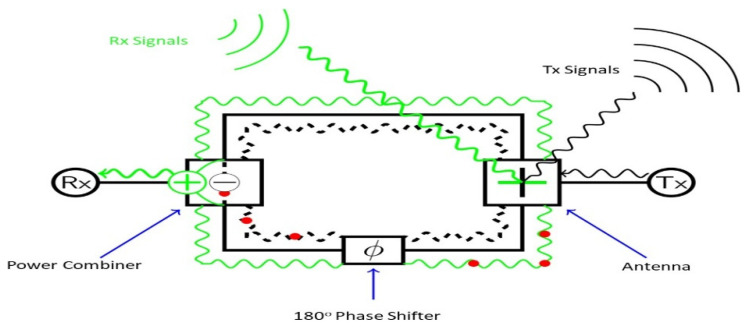
Proposed SIW antenna system operation (red dot represent the out-of-phase component) [21].

**Figure 11 micromachines-13-00717-f011:**
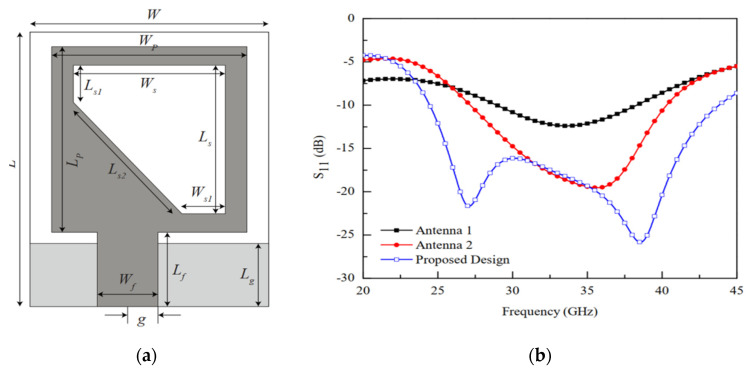
Proposed patch antenna with triangular slot loaded. (**a**) Top view. (**b**) Enhancement in bandwidth due to slot in blue [31].

**Figure 12 micromachines-13-00717-f012:**
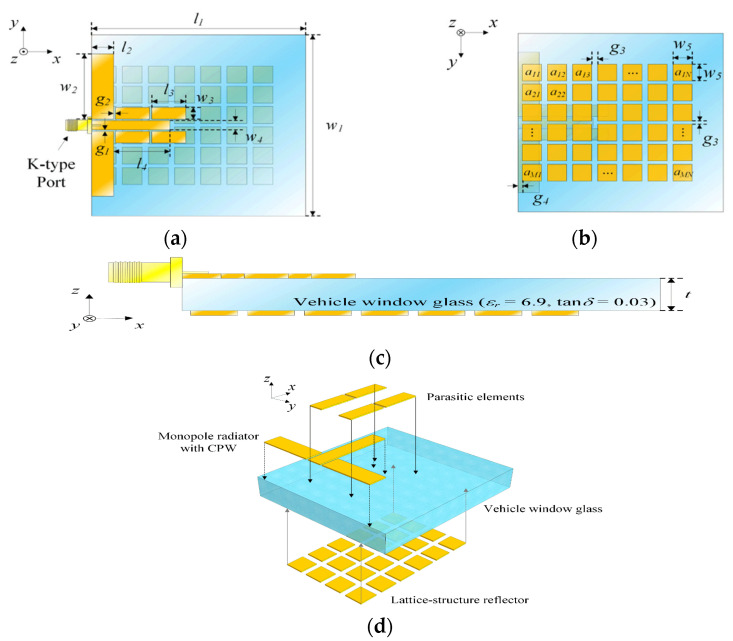
Geometry of the proposed printed antenna: (**a**) top view; (**b**) bottom view; (**c**) side view; (**d**) isometric view [28].

**Figure 13 micromachines-13-00717-f013:**
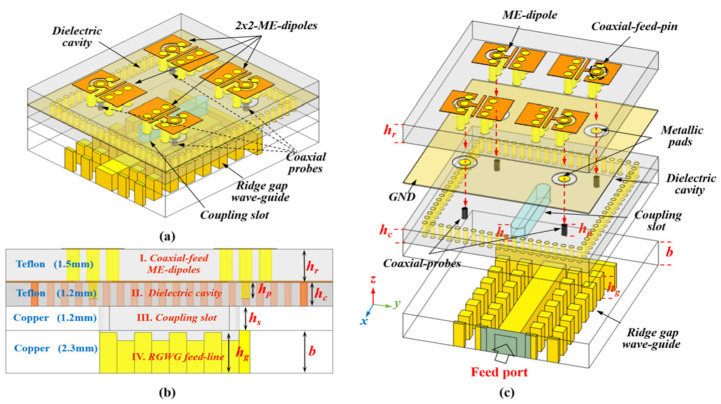
Hybrid feed geometry of a 2 × 2 unit for ME dipole subarray. (**a**) Perspective view. (**b**) Side view. (**c**) Exploded view [57].

**Figure 14 micromachines-13-00717-f014:**
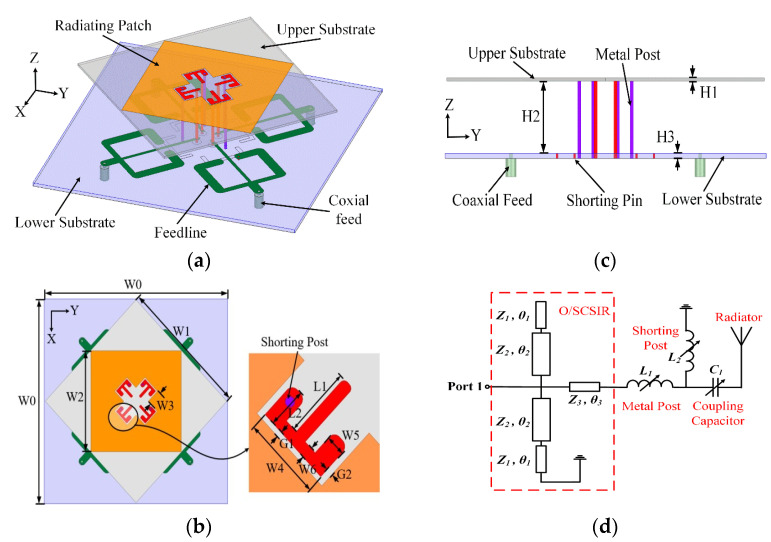
Antenna configuration: (**a**) 3D view, (**b**) top view, (**c**) front view, (**d**) Equivalent circuit [39].

**Figure 15 micromachines-13-00717-f015:**
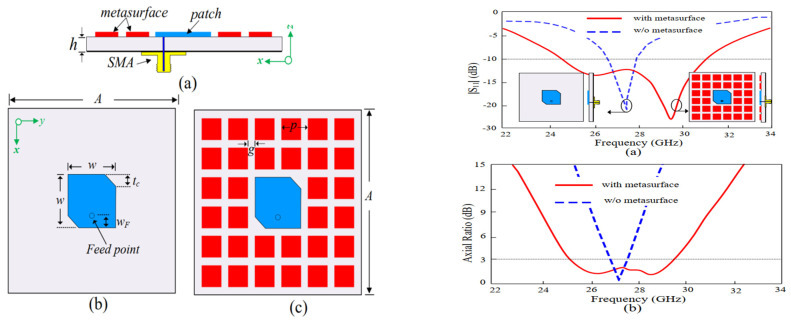
Antenna geometry: (**a**, **left**) side view (**b**,**c**, **left**) top and bottom view, and (**a**,**b**) (**right**) impact of MS on S_11_ and axial ratio [61].

**Figure 16 micromachines-13-00717-f016:**
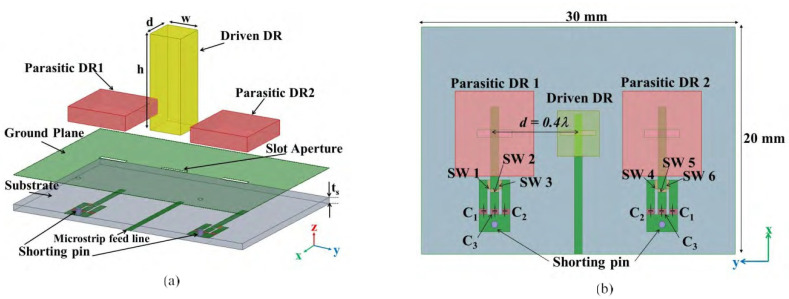
The geometrical configuration of the proposed DRA array. (**a**) 3D view. (**b**) Top view [62].

**Figure 17 micromachines-13-00717-f017:**
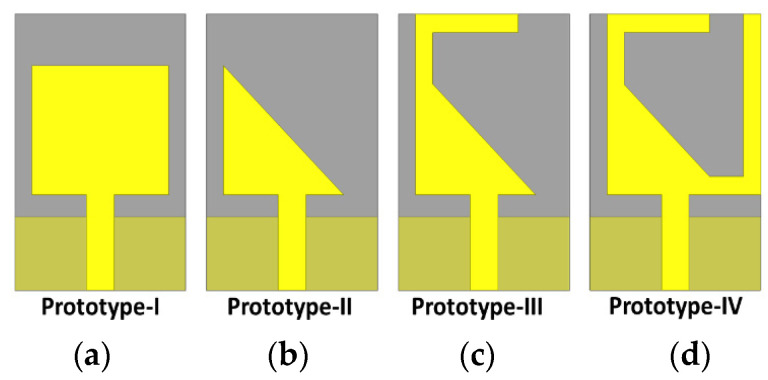
Evolution steps of antenna presented in Figure 17. Different geometrical shapes (**a**–**d**).

**Figure 18 micromachines-13-00717-f018:**
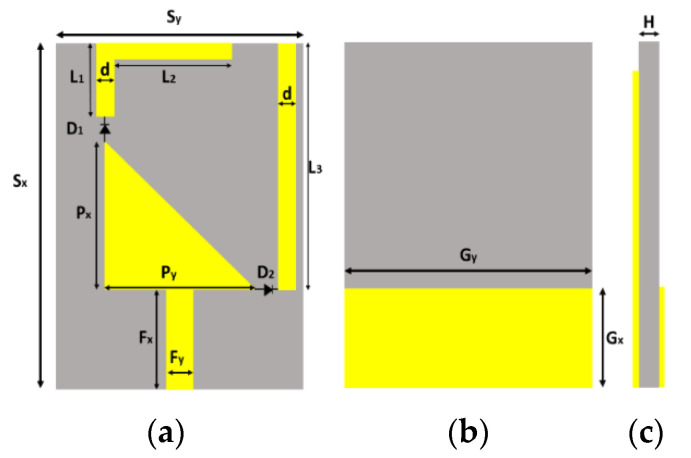
Final geometry of antenna element. (**a**) Top view, (**b**) bottom view, and (**c**) side view [73].

**Figure 19 micromachines-13-00717-f019:**
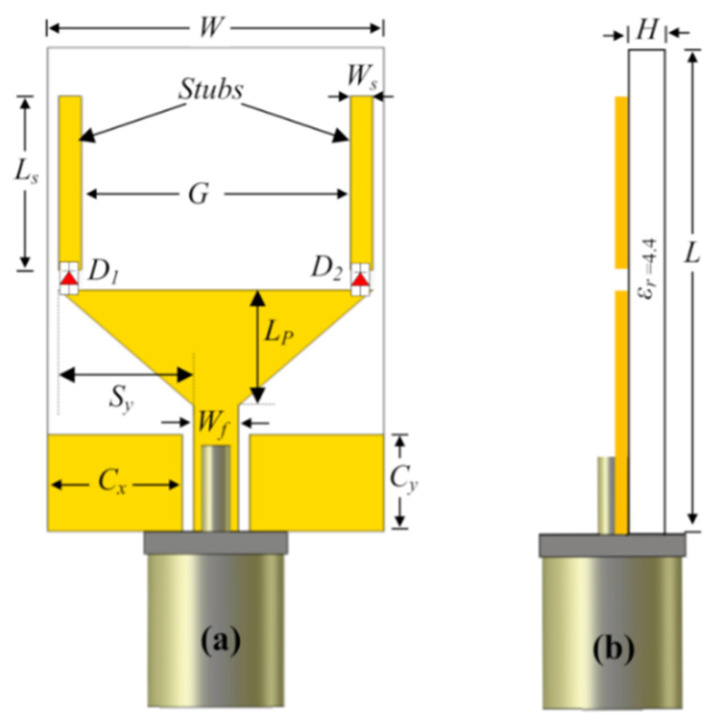
A reconfigurable antenna [33]. (**a**) Front view and (**b**) side view.

**Figure 20 micromachines-13-00717-f020:**
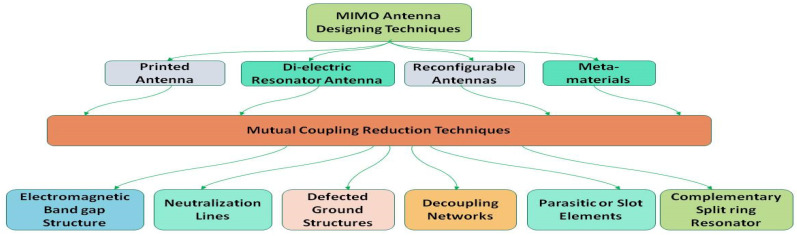
Designing and mutual coupling reduction technique of MIMO antennas, a summarized generic view.

**Table 1 micromachines-13-00717-t001:** Performance Enhancement Techniques: Advantages and Disadvantages.

Reference Antennas	Performance Enhancement Techniques	Advantages	Disadvantages
[45,46]	Substrate Choice	Substrate with low permittivity characteristics have a significant impact on the performance of an antenna and provide enhanced gain, wide bandwidth and high efficiency. Having high permittivity characteristics improves the value of return loss.	Substrate with low permittivity is costly and not easily available.
[47,48,49]	Mutual Coupling Reduction /Decoupling	Excellently improves the impedance matching and directly enhances both the gain and efficiency. Mutual coupling reduction techniques also reduce the size of antenna.	Mutual reduction has an impact on antenna designing and increases the complexity.
[50,51]	Multielement	It significantly improves the return loss, bandwidth and radiation efficiency, besides these properties, it also effectively reduces the side- and back-lobe levels.	For such techniques, the feeding network is a difficult task to design and makes complexity to some extent.
[52,53,54]	Corrugation	Improvement in the gain, efficiency, and bandwidth as well as return loss.	Significantly reduces the input impedance.
[55,56]	Dielectric Lens	Gain enhancement improvement in front-to-back ratio with stable radiation pattern. Enhancement in the gain, improvement in the front-to-back ration, stability in the radiation pattern, and radiation in the front-side direction.	The size of antenna definitely increases.

**Table 2 micromachines-13-00717-t002:** A Summary of different Antennas Performances with Different Fabrication Techniques.

References Antennas	Number of Unit Cells		Total Size Array (λ0)		Effective Bandwidth (S11 < −10 dB)		Maximum Gain		Radiation Efficiency (%)	Fabrication Techniques
[65]	8 × 8	150 × 75 × 7	6.4 × 6.4 × 0.17	Ground Slot	34–40.1 Hz	3.3–6.0	24 dBi	>18	41%	PCB
[66]	5 × 6	124 × 74 × 6	5.07 × 3.82 × 0.1	Ground Slot	33.95–34.86 GHz	3.3–3.6	17.09 dBi	>15	22%	PCB
[67]	8 × 8	150 × 80 × 0.8	6.12 × 6.8 × 0.5	No	55.4–66.5 GHz	3.4–3.6	26.1 dBi	>17.5	70%	Multi-Layer PCB
[68]	4 × 4	150 × 75 × 0.8	2 × 2 × 0.12	Orthogonal Polarization	25.5–40.2 GHz	3.3–3.8	16.1 dBi	>15	83%	LTCC
[69]	16 × 16	150 × 75 × 8	15.7 × 16 × 0.8	Orthogonal Polarization	71–86 GHz	3.4–3.6	32.9 dBi	>17	86.60%	Diffusion Bonding
[70]	4 × 4	145 × 75 × 6	5.3 × 5.3 × 1.1	No	29.6–30.7 GHz	3.4–3.6	22.4 dBi	>15	99%	Machining
[71]	4 × 4	150 × 75 × 7	3.5 × 3.4 × 0.3	No	28.8–34 GHz	3.4–3.6	21.2 dBi	>12.7	70%	PCB
[72]	4 × 8	150 × 75 × 7	11.8 × 11.4 × 2.2	Neutralization Line	86.7–102.2 GHz	3.4–3.6	23 dBi	>11.5	N/A%	PCB + Machining
[58]	8 × 8	140 × 70 × 1	5.4 × 5.4 × 0.6	No	26.05–31.15 GHz	3.4–3.6	25 dBi	>11.2	85%	
						3.4–3.8		>15.5		

**Table 3 micromachines-13-00717-t003:** Comparison of different antennas using pin diode as a witching component.

References Antennas	Antenna Size (λ0 × λ0)	Number of Pin Diodes	Single Band	Multi Band	Wide Band	Bandwidth (%)
[75]	1.00 × 0.41	2	4	4	8	26.4%, 37.4%
[76]	0.62 × 0.41	1	4	4	8	13.5%, 35.72%, 9.94%
[77]	0.36 × 0.33	6	4	8	4	123.5%, 28.5%
[78]	0.60 × 0.28	2	4	8	4	44.89%, 10.55%
[79]	0.55 × 0.59	4	8	4	4	74%, 8.2%, 9.79%, 15.4%
[33]	150 × 0.16	2	4	4	4	64.40%, 24%, 25.5%

**Table 4 micromachines-13-00717-t004:** A comparison of different switch components [83,84,85,86,87].

Switching Types	Supremacy/ Advantages	Weaknesses/ Disadvantages
PIN Diode	Extremely reliable Very low in cost Most probable choice for reconfiguration	Power handling capability is high The tuning speed is very high High DC biasing capability in ON state
MEMS	Feasible for small flow of current continuous tuning Integration is easy	A nonlinear source Lower range dynamically Biasing circuitry is complex
Varacters	Linear impedance bandwidth with high isolation Comparatively less noise figure and low power losses	Voltage control level is high. The switching speed is slow Limited life cycle is limited compared with other components.

**Table 5 micromachines-13-00717-t005:** Mutual Coupling Reduction/Decoupling Techniques Advantages and Disadvantages.

Reference Antennas	Performance Enhancement Techniques	Advantages	Disadvantages
[90]	Neutralization Lines	This technique is mostly used in compact-size antenna to decouple the closed interelement coupling.	Complexities in structure
[91,92]	Dielectric Resonator Antenna	Bandwidth Gain and Efficiency Enhancement.	Complexities in structure
[93,94]	Neutralization Lines	This technique is mostly used in compact-size antenna to decouple the closed interelement coupling.	Complexities in structure
[95,96]	Frequency Reconfigurable Antenna	It provides feasibility for antenna to be in compact size and excellent provision for compactness dimensionally.	External component provision is not an easy task.
[97,98,99,100]	Slot or Parasitic element Metamaterials Decoupling Network	Diversity gain, bandwidth, and efficiency could excellently be improved. Enhancement in the diversity gain, bandwidth and envelop correlation coefficient (ECC) Diversity gain and impedance matching could be improved effectively.	Designing and decision about position is difficult and a time-consuming activity. Decisions about position and designing are not an easy job. Complexity in design as well as low in gain

**Table 6 micromachines-13-00717-t006:** Comparison of Different Techniques employed for Mutual Coupling Reduction/Decoupling.

Reference Antennas	Geometry	Decoupling Techniques	Effective Bandwidth (GHz)	Isolation(dB)	Channel Capacity Loss	ECC
[101]	Non Planar	Decoupling based on FSS	3–11	20	<0.20	<0.20
[102]	Non Planar	Parasitic structure for decoupling	3.1–10.6	20	<0.70	<0.1
[103]	Non Planar	Configuration based on 3D element distance	3–11	20	-	<0.5
[104]	Planar	Decoupling structure based on grounded slits	2–11	15	-	<0.2
[89]	Planar	Orthogonal polarization	2.5–2.7	12	-	<0.12
[105]	Planar	Decoupling structure of ENG-NZI metamaterial	3.4–3.6, 4.8–5.0	12	<0.08	<0.15
[106]	Planar	DN (secoupling network)	3.4–3.6	14	-	<0.2
[107]	Planar	Element positioning + geometrical slotting	3.27–5.92	14.5	-	<0.1
[108]	Planar	Port distance plus positioning of elements	3.2–6.1	18	-	<0.21
[109]	Planar	Parasitic structure for decoupling	3–11	20	<0.35	<0.0025
[110]	Planar	Decoupling structure of ENG-NZI metamaterial	3–11	28	<0.30	<0.1

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
