# Peer review of "Latest Performance Improvement Strategies and Techniques Used in 5G Antenna Designing Technology, a Comprehensive Study"

_micromachines, 2022, doi:10.3390/mi13050717_

Round 1

Author Response

Reviewer 1 Comments:

  1. In Figure 7, a common aperture 5G antenna for integration of sub 6 GHz and mm-Wave is shown. The sub 6 GHz and mm-Wave can also be integrated with other techniques such as reconfigurability [R1, "Integration of Sub-6-GHz and mm-Wave Bands With a Large Frequency Ratio for Future 5G MIMO Applications," in IEEE Access, vol. 9, pp. 11241-11251, 2021, DOI: 10.1109/ACCESS.2021.3051066.] and low pass filter [R2, "Low-Pass Filter Based Integrated 5G Smartphone Antenna for Sub-6-GHz and mm-Wave Bands," in IEEE Transactions on Antennas and Propagation, vol. 69, no. 9, pp. 5424-5436, Sept. 2021, DOI: 10.1109/TAP.2021.3061012.]. I would suggest including this section as well.

Note:The revised sections for reviewer 1.are highlited in yellow

Response 1. Reviewer 1.

As per guidance of the reviewer 1,I have Proper included the detail of both referenced work on Page 6,7 of the paper and have properly referenced and cited the said work with reference no.14,15.which realy improved the quality of the paper.

  1. In 5. Reconfigurability, both frequency reconfigurability [R3, https://doi.org/10.1016/j.aeue.2018.10.012 ] and pattern reconfigurability [R4, "A Pattern Reconfigurable Antenna Design for 5G Communication System," 2021 International Symposium on Antennas and Propagation (ISAP), 2021, pp. 1-2, DOI: 10.23919/ISAP47258.2021.9614435.] should be discussed.

Response 2. Reviewer 1.

Similarly to reference1,2  per instructions of reviewer both referenced work has been included to the reconfigurability section  on page 16,17 and are referenced with reference no.40 and 53.

  1. In 3.11 Performance enhancement using Metamaterials: the author should also discuss the performance enhancement by using the EBG structures [R5, Development of 60-GHz millimeter wave, electromagnetic bandgap ground planes for multiple-input multiple-output antenna applications. Sci Rep 10, 8541 (2020). https://doi.org/10.1038/s41598-020-65622-9 ].

Response 3. Reviewer 1.

  As per instruction of Reviwer 1 ,after styding the referenced work which was more          relevant to the Sub section EBG structure,so added the Ref 5 work to EBG section and is cited and referenced properly  page no.18 with reference no.[120]

  1. The quality of the figures should be improved.

Response 4. Reviewer 1.

     The Quality of the Figures  is much improved in the Revised version.

     As in the previous version the quality of the figures no.11,14 and 15 was poor.

  1. The paper contains many grammatical errors and should be revised properly.

                I have rechecked the paper for grametical errors and are removed in the revised version.

Thank you very much Sir for your professional approach of guidance which really helped me to improve the quality of my Paper.

Reviewer 2 Report

The manuscript presents a comprehensive survey on the performance improvement strategies and techniques used in 5G antenna designing technology. Though the investigation is well researched, a major revision is necessary for the manuscript to be accepted for publication.

  1. The abstract needs to be re-written to clearly highlight the target achievements of the investigation. Just stating that "The survey of possible bandwidth enhancements is presented in this work" in the last sentence of the abstract is not acceptable.
  2. The manuscript is not written in a way that follows a logical flow of information. For example, comparison tables are included in no particular order. The authors should revised the manuscript and include related comparison tables in each relevant section.
  3. The authors should include a section on the future challenges in the field of antenna design, and suggest possible remedies to such issues.
  4. Similar to the abstract, the conclusion should be completely revamped to demonstrate the achievements of the survey. The conclusion, like the abstract, is always the first point of contact with the wider reading audience and should be written in a way that attracts the reader to read the whole article.  

Author Response

Reviewer 2 Comments:

The manuscript presents a comprehensive survey on the performance improvement strategies and techniques used in 5G antenna designing technology. Though the investigation is well researched, a major revision is necessary for the manuscript to be accepted for publication.

  1. The abstract needs to be re-written to clearly highlight the target achievements of the investigation. Just stating that "The survey of possible bandwidth enhancements is presented in this work" in the last sentence of the abstract is not acceptable.

Note:The revised sections for reviewer 1.are highlited in green

Response 1. Reviewer 2.

As per instructions of the Reviewer 2, the Abstract of the Paper is properly revised which clearly highlights the Target Achievements and Investigations being reviewed in the literature and have really improved the quality of the paper.

  1. The manuscript is not written in a way that follows a logical flow of information. For example, comparison tables are included in no particular order. The authors should revised the manuscript and include related comparison tables in each relevant section.

Response 2. Reviewer 2.

The manuscript is rearranged in a proper way to be presented in a logical way,as some of the tables were not in an appropriate order for examle the Table1,2 was placed together .in the previous version of paper. the table two is shifted separately to the last of the relevant section of different Techniques  and now it is Title  as table 1 on p.15.the tale 1  is now placed as table 2 which was missing with Title and relevant detail. In revised version of 4.2 with proper title and its detail  it is shown as  table no.2 which was updated with proper title and summerized proper explanation.

  1. The authors should include a section on the future challenges in the field of antenna design, and suggest possible remedies to such issues.

Response 3. Reviewer 2.

As per instructions of the Author  a separate section on future challenges in the field of antenna design  is included  as a separate section  no.7 ,Page no 21,22.is in the revised file of the paper.

  1. Similar to the abstract, the conclusion should be completely revamped to demonstrate the achievements of the survey. The conclusion, like the abstract, is always the first point of contact with the wider reading audience and should be written in a way that attracts the reader to read the whole article. 

Response 4. Reviewer 2.

Similar to Abstract the Conclusion is also updated covering the deficiencies of the    information related to conclusion on page. 22.

Thank you very much Sir for your professional approach of guidance which really helped me to improve the quality of my Paper.

Round 2

Reviewer 1 Report

The authors properly discuss all of my concerns. Now, the paper can be published in its current form.